# Functional Characterization of a Familial ALS-Associated Missense *TBK1* (p-Arg573Gly) Mutation in Patient-Derived Lymphoblasts

**DOI:** 10.3390/ijms24032847

**Published:** 2023-02-02

**Authors:** Gracia Porras, Silvana Ruiz, Inés Maestro, Daniel Borrego-Hernández, Alberto G. Redondo, Ana Martínez, Ángeles Martín-Requero

**Affiliations:** 1Department of Molecular Biomedicine, Centro de Investigaciones Biológicas, Margarita Salas (CSIC), Ramiro de Maeztu 9, 28040 Madrid, Spain; 2Department of Structural and Chemical Biology, Centro de Investigaciones Biológicas, Margarita Salas (CSIC), Ramiro de Maeztu 9, 28040 Madrid, Spain; 3ALS Research Lab, Hospital 12 de Octubre Research Institute (i+12), 28041 Madrid, Spain; 4Centro de Investigación Biomédica en Red de Enfermedades Neurodegenerativas (CIBERNED), Instituto de Salud Carlos III, 28031 Madrid, Spain

**Keywords:** amyotrophic lateral sclerosis, TBK1, RIPK1, TDP-43, lymphoblasts

## Abstract

The goal of this work was to elucidate the pathogenic mechanism of an ALS-associated missense mutation, p.Arg573Gly (R573G), in the *TBK1* gene. In particular, we seek to analyze the influence of this variant on the cellular levels and the function of TBK1 in immortalized cells from an ALS patient. The patient (Code# E7) belonged to a Spanish family with autosomal dominant disease manifesting in the sixth decade as either dementia or ALS. Four control individuals without signs of neurological disease were also included in this study. Our results indicate that the R375G *TBK1* mutation did not affect the levels of mRNA nor the total TBK1 content; however, we observed a significant decrease in the levels of TBK1 phosphorylation, which is essential for TBK1 activity, as well as a significant reduction in the phosphorylation of p62 and RIPK1, known substrates for TBK1. Lymphoblasts from the R573G *TBK1* mutation carrier patient display pathological TDP-43 homeostasis, showing elevated levels of phosphorylated TDP-43 and accumulation of the protein in the cytosolic compartment. In addition, the functional decrease in TBK1 activity observed in the E7 patient did not alter the autophagy flux, but it seems to be enough to increase ROS levels as well as the expression of pro-inflammatory cytokine IL-6.

## 1. Introduction

Amyotrophic lateral sclerosis (ALS) is a neurodegenerative disorder characterized by adult-onset loss of motor neurons and destruction of neuromuscular junctions, leading to progressive muscle weakness and atrophy [1,2]. The mean age of ALS onset varies from 50 to 65 years, with a median age of onset of 64 years old [1]. Death usually occurs within 3 to 5 years from the symptom onset, commonly caused by respiratory failure.

Approximately 10% of amyotrophic lateral sclerosis (ALS) cases have a family history [3]. Four genes, *SOD1*, *C9ORF72*, *TARDBP*, and *FUS*, account for over 50% of the familial forms of ALS [4]. Mutations in *TBK1* (TANK binding kinase 1) have been linked with ALS [5]. Mutations in this gene has been found in approximately 4% of ALS and frontotemporal dementia (FTD) patients [6]. This gene codes for a protein kinase involved in the immune response and autophagy, among other pathways [7].

TBK1 is a 729 amino acid protein that contains four functionally distinct domains: a kinase domain, responsible for its kinetic activity; a ubiquitin-like domain; a scaffold dimerization domain; and a C-terminal domain, involved in TBK1 association with binding partners such as optineurin or p62, which are important autophagy receptors [8]. It has been reported that both TBK1 homodimerization and autophosphorylation at the serine 172 residue are necessary for its activation [9,10]. Mutations affecting all four domains have been documented [7,11]. Most of the ALS-linked *TBK1* mutations generate premature stop codons, leading to nonsense-mediated mRNA decay and haploinsufficiency. However, missense mutations have also been found in ALS patients, leading to functional loss of the kinase activity [12].

The aim of this work was to elucidate the pathogenic mechanism of a *TBK1* missense mutation (R573G) located in the dimerization domain [11] by analyzing the influence of this variant on cellular levels and the function of TBK1 in immortalized lymphocytes from an ALS patient. The patient (Code# E7) belonged to a Spanish family with autosomal dominant disease manifesting in the sixth decade as either dementia or ALS [13]. The expression levels and kinase activity of TBK1, together with the TDP-43 pathology, autophagic flux, and expression levels of pro-inflammatory cytokines, were analyzed against control lymphoblasts. We have used lymphoblastoid cell lines, based on previous reports from this laboratory indicating that lymphoblasts from ALS patients recapitulate main pathological hallmarks of the disease [14,15]. Moreover, compelling evidence indicates that ALS can be considered as a multisystem degeneration disorder, affecting cells other than motor neurons [16]. Interestingly, non-neuronal cells, such as astrocytes, microglia, skeletal muscle fibers, and peripheral blood mononuclear cells, may contribute to motor neuron degeneration [17]. Therefore, the use of peripheral cells from ALS patients provides a useful model to gain insight into ALS pathogenesis.

We report here that the R573G *TBK1* mutation impaired the kinase activity of the protein without changes in either mRNA or protein levels. Lymphoblasts from the E7 patient (carrier of R573G *TBK1* mutation) display TDP-43 pathological features, i.e., increased phosphorylation, cleavage, and cytoplasmic TDP-43 accumulation, together with enhanced ROS levels and increased expression of IL-6.

## 2. Results

### 2.1. Influence of R573G TBK1 Mutation on Expression Levels and Activity of TBK1

We first studied whether R573G *TBK1* mutation had any effect on TBK1 expression by determining the TBK1 mRNA levels by q-RT-PCR and the cellular content of TBK1 protein by Western blotting. To this end, mRNA and protein extracts were prepared from lymphoblasts derived from the ALS patient carrier of the R573G mutation (E7), and from a pool of lymphoblasts derived from four healthy individuals. We found neither differences in mRNA TBK1 nor in protein levels in lymphoblasts derived from the E7 patient as compared with control cells (Figure 1A). Both mRNA and protein levels of pooled cells were similar to the mean value of determinations made in individual control cell lines (Appendix A).

TBK1 activity depends on the autophosphorylation of the molecule at Ser172 residue. For this reason, we determined the influence of *TBK1* mutation on the levels of phosphorylation of TBK1 using a phospho-specific antibody. As shown in Figure 1B a significant reduction in TBK1 phosphorylation was found in lymphoblasts from the E7 patient. Moreover, we observed a decrease in the levels of phosphorylated p62 and RIPK1, well known TBK1 substrates [18,19], suggesting that the *R573G TBK1* mutation is responsible for the decrease in the kinase activity of the protein (Figure 1B).

### 2.2. Cell Viability and ROS Generation in R573G Mutant Lymphoblasts

Lymphoblasts from the mutant ALS patient showed a slight decrease in cell viability, as indicated in Figure 2A. As expected, the withdrawal of serum from the medium resulted in a reduction in the number of viable cells (Figure 2A, right panel). Under these conditions, there was a significant decrease in viability of E7 lymphoblasts compared with that observed in control cells. In addition, we determined the ROS levels in control and E7 lymphoblasts measured using the redox sensitive fluorescent probe, CM-H2DCFDA. As it can be observed in Figure 2B, there was an increase in ROS levels in lymphoblasts derived from the E7 patient when compared with those found in control subjects.

### 2.3. R573G TBK1 Mutation Alters TDP-43 Homeostasis

Since post-transduction alterations such as increased phosphorylation and cleavage of the TDP-43 protein have been found in 97% of ALS patients [20], we then evaluated the effects of R573G *TBK1* mutation on the levels and phosphorylation status of TDP-43 in control and E7 derived lymphoblasts. TDP-43 phosphorylation was assessed by Western blotting using a phospho-specific (Ser409/410) anti-TDP-43 antibody, while cellular levels of TDP-43 were detected with an anti N-terminal TDP-43 antibody. Figure 3 indicates that lymphoblasts from the E7 patient had higher levels of phosphorylated full-length (43 KDa) and truncated TDP-43 (35 and 25 KDa fragments) than the control cells. These observations are in line with previous reports showing alterations in TDP-43 phosphorylation in lymphoblasts from sporadic ALS patients [14,15]. In contrast, we did not find differences in total TDP-43 levels between control or E7 cells.

Cytosolic TDP-43 accumulation has been observed in most ALS cases [20]. Thus, we addressed the issue as to whether R573G *TBK1* mutation perturbs the balance between nuclear and cytosolic content of TDP-43 protein. For this purpose, we performed nuclear and cytoplasmic fractionation and analyzed the protein extracts by Western blot analysis. Figure 4 shows higher levels of cytosolic TDP-43, together with a clear reduction in nuclear TDP-43 content, in cells derived from the E7 patient compared to control cells.

The influence of the R573G *TBK1* mutation on TDP-43 subcellular localization was further assessed by immunocytochemistry and confocal laser microscopy (Figure 5). Quantification of TDP-43 fluorescence clearly indicates that cytosolic levels of TDP-43 are increased in mutant cells.

### 2.4. Influence of R573G TBK1 Mutation on Autophagic Flux and Pro-Inflammatory Cytokine Levels

Protein aggregates and defective mitochondria are documented in ALS. As autophagy is involved in both clearance of protein inclusions and damaged mitochondria, we seek to carry out a comparative analysis of autophagic flux in lymphoblasts from the E7 patient and control individuals. To this end, we determined the levels of the autophagy marker LC3-II in the absence or in the presence of the autophagy blocker hydroxychloroquine (HCQ) [21]. As shown in Figure 6A, the autophagic flux, assessed by the ratio of LC3-II levels with and without HCQ, appears to be similar in both the E7 lymphoblast and control cells. Next, we evaluated the mRNA expression levels of pro-inflammatory cytokines IL1B, IL-6, and TNFα in control and mutant lymphoblasts, as the role of TBK1/RIPK1 axis is well known in neuroinflammation [22]. Figure 6B shows a significant increase in IL-6 in lymphoblasts derived from the R573G *TBK1* mutation carrier.

## 3. Discussion

In this work, we have studied the functional impact of a missense *TBK1* mutation (R375G) in lymphoblasts derived from an ALS patient. This patient belongs to a family with several members affected by motor neuron or dementia diseases.

To date, several mutations in the *TBK1* gene have been reported. Most of the ALS-linked *TBK1* mutations are nonsense that generate premature stop codons leading to haploinsufficiency. There are also missense *TBK1* mutations, the pathogenicity and mechanisms of which are unclear.

The R375G *TBK1* variant found in heterozygosis in the E7 patient is a missense mutation located in the dimerization domain [11]. Our results indicate that the R375G *TBK1* mutation did not affect the levels of mRNA nor the total TBK1 content, as quantification of the Western blots confirmed similar levels of TBK1 protein in both control- and ALS patient-derived lymphoblasts. However, by using a *p*-S172-TBK1 antibody, we observed a significant decrease in the levels of TBK1 phosphorylation, which is essential for TBK1 activity, and therefore might account for its pathogenicity.

Moreover, the levels of phosphorylated p62 and RIPK1, well-known TBK1 substrates [18,19], were also reduced in the R375G *TBK1* mutation carrier. These observations support the hypothesis that the *TBK1* mutation linked to the disease may impair its auto phosphorylation, perturbing its capacity to phosphorylate multiple substrates [10]. It is not known at present whether decreased phosphorylation of TBK1 protein results in altered self-interaction or impaired interaction with other kinases.

Lymphoblasts from the R573G *TBK1* mutation carrier patient show changes in cell viability, which are more evident in the absence of serum in the culture medium. We observed a significant increase in ROS generation between the control- and patient-derived lymphoblasts. Increased levels of ROS have been widely reported in ALS [23]. We previously reported enhanced ROS generation in lymphoblasts from sporadic ALS patients [24]. Moreover, increased ROS levels were reported in lymphoblasts of familial ALS cases with SOD1 mutations [25] and fibroblasts of patients with C9orf72 G4C2 repeat expansions [26].

In accordance with the fact that TDP-43 pathology is present in 97% of ALS patients (both sporadic and familial cases) [27], we found a significant increase in TDP-43 phosphorylation of the full-length protein as well as the 35KD C-terminal fragment. Moreover, we observed higher accumulation of TDP-43 protein in the cytosolic compartment. These observations are identical to changes in TDP-43 homeostasis reported previously in lymphoblasts from sporadic ALS patients [14], as well as in cells derived from FTLD-TDP patients [28], reinforcing the idea that peripheral cells from patients recapitulate hallmarks of affected neurons. In addition, it is worth mentioning that accumulation of TDP-43 in the cytoplasm has also been documented in non-transformed circulating lymphomonocytes from patients [29].

It has previously been shown that deletion of TBK1 impaired autophagy and accelerated disease progression in a murine model of ALS [30]. Moreover, alterations in autophagy had been documented in ALS cases harboring nonsense *TBK1* mutations leading to haploinsufficiency [6], but the role played by missense TBK1 mutations on autophagy is less clear. Our results show that a functional decrease in TBK1 activity without changes in TBK1 protein levels is not sufficient to impair autophagy flux. It is believed that phosphorylation of p62 at Ser 403 promotes p62 binding to ubiquitin, enhancing autophagic activity [18]. Thus, the reduced phosphorylated p62 levels found in R573G *TBK1* mutant lymphoblasts are consistent with the lack of induction of autophagy flux, which might leave the cells more susceptible to stress or injury. In addition, the inability to promote TDP-43 autophagic degradation may explain the presence of TDP-43 aggregates in R573G *TBK1* carriers. Nonetheless, autophagy is a challenging process to be studied in further analysis, including of different *TBK1* variants associated with ALS.

TBK1 has been classically considered as an activator of innate immunity. TBK1 acts upstream of type-I-interferon (IFN-I) and inflammatory cytokine production via NF-κB [31]. However, TBK1 is also capable of negatively regulating inflammatory TNF-α-mediated cell death via an inactivating interaction with RIPK1 [32], and it has been suggested that a reduced inhibition of RIPK1 in the CNS might cause a predisposition to neuroinflammation and neurodegeneration [33]. Our finding that R573G *TBK1* mutant lymphoblasts display reduced levels of phosphorylated (inactive) RIPK1 prompted us to analyze the expression levels of inflammatory cytokines. Analysis of the mRNA expression levels of the pro-inflammatory cytokines IL-1β, IL-6, and TNF-α in the lymphoblasts derived from the R573F *TBK1* mutant carrier showed a significant increase in the levels of IL-6 in comparison with levels found in control individuals. Our results are in line with previous work indicating that TBK1 deficiency induces the expression of pro-inflammatory cytokines in bone marrow-derived macrophages and neutrophils [34]. Moreover, increased levels of plasma IL-6 have been reported to correlate with disease progression and muscle weakness [35], and there is also an association between higher IL-6 levels and respiratory function in ALS [36].

Altogether, our results demonstrated that the missense *TBK1* mutation (R573G) significantly reduced the activity of the TBK1 protein, leading to impaired TDP-43 homeostasis, increased ROS generation, reduced cell viability, and elevated levels of the pro-inflammatory cytokine IL-6, most likely due to the TBK1 deficiency-induced activation of RIPK1. It is suggested that therapeutic strategies aiming to reduce ROS levels and RIPK1 activity, as well as to recover TDP-43 homeostasis, could be useful for ALS patients carrying a dysfunctional TBK1 protein.

## 4. Materials and Methods

### 4.1. Materials

RPMI 1640 culture medium was obtained from Biowest/Labclinics (Barcelona, Spain), fetal bovine serum (FBS) was purchased from Merck (Madrid, Spain). PVDF (polyvinylidene difluoride) membranes for Western blots were purchased from Bio-Rad (Richmond, CA, USA). The enhanced chemiluminiscence (ECL) system was from Amersham (Uppsala, Sweden). All other reagents were of molecular grade. Antibodies against human TDP-43 (Cat#: 10782-2-AP) and anti-phospho (Ser409/410)-TDP-43 (Cat#: 22309-1AP), anti-RIPK1 (Cat#: 17519-1-AP), anti-phosho (Ser161)-RIPK1 (Cat# 66854-1-Ig), and anti-TBK1 (Cat#: 67211-1-Ig) were obtained from Proteintech (Manchester, UK). Antibodies anti-SQSTM1/p62b (Cat#: GTX111393) and anti-phospho (Ser403)-p62 (Cat#: GTX128171) were obtained from GenenTex Inc (Irvine, CA, USA). Antibodies β-actin (sc-81178) and α-tubulin (sc-23948) were obtained from Santa Cruz Biotechnologies-Biogen (Madrid, Spain) and anti-Lamin B1 (Cat#: NA12) and anti phospho-(Ser172)-TBK1 (NAK (Cat #:D52C2)) were purchased from Calbiochem (San Diego, CA, USA). Anti-LC3 (Cat# L7543) was from Sigma-Aldrich (Alcobendas, Spain).

Source of cell lines: Epstein–Barr virus (EBV)-immortalized cell lymphocytes were from one ALS patient with a heterozygote R573G *TBK1* mutation, who was a male, Caucasian, diagnosed at 69 years old, and died at 75 years old; and from a control pool of four healthy individuals from the cell lines in our repository. Demographic and clinical information are presented in Table 1.

Participants or their relatives gave written informed consent. This study was approved by the Hospital Doce de Octubre and the Spanish Council of Higher Research Institutional Review Boards. Case E7 was diagnosed by applying the revised El Escorial criteria [37]. Genetic testing was performed analyzing a panel including 48 ALS-related genes. Only the R573G *TBK1* variant was found. The Arg573Gly change is located in the dimerization domain of the protein.

### 4.2. Culture of Human Lymphoblasts

Cells were grown in suspension vertically in T flaks, in approximately 10 mL of RPMI-160 (Gibco, RBL) medium that contained 2 mM L-glutamine, 100 mg/mL penicillin/streptomycin, and 10% (*v*/*v*) fetal bovine serum (FBS) and maintained in a humidified 5% CO_2_ incubator at 37 °C. The medium was changed routinely every two days my removing that medium above the settled cells and substituting it with an equal volume of fresh medium.

### 4.3. Determination of Cell Viability

Cell viability was determined through an MTT, 3-(4,5-dimethylthiazol-2-yl)-2,5-diphenyltetrazolium bromide, assay. This assay measures the ability of mitochondria to convert MTT into insoluble formazan crystals, the quantity produced is directly proportional to the number of cells alive [38]. Cells (1 × 10^6^ cells × mL^−1^) were incubated with 0.25 μg/μL of MTT in a reaction volume of 200 μL. After incubation, DMSO was added to dissolve formazan crystals. Dye absorbance in viable cells was measured at 590 nm with 620 nm as a reference wavelength in a Varioskan Flash spectrophotometer (Thermo Scientific Laboratories, Rockford, IL, USA).

### 4.4. Analysis of mRNA Levels by Quantitative Real-Time PCR (q-RT-PCR)

RNA was extracted from cultured cells using Trizol ™ agent (Invitrogen, Waltham, MA, USA). The yield of the RNA extraction was quantified spectrophotometrically in a NanoDrop and its quality was determined using the ratios A260/A280 and A260/A230. The RNA was treated with DNAase I Amplification Grade (Invitrogen). An amount of 1 µg of RNA was reverse-transcribed using the Superscript III reverse transcriptase kit (Invitrogen). Subsequently, a quantitative real-time PCR was carried out in triplicate, using TaqMan Universal PCR MasterMix No Amperase UNG (Applied Biosystems, San Francisco, CA, USA) reagent according to the protocol described by the manufacturer.

Primers were designed using the Universal Probe Library for Humans (Roche Applied Science, Penzberg, Germany). The final concentration of the primers was 20 µM. The sequences of the Forward and Reverse primers were as follows: for TBK1 (forward and reverse) 5′-AGAAGAATATTTGCACCCTG-3′ and 5′-AAAATGTTACCCCAATGCTC-3; and 5′-CCATTATCCCCAGCAAAAAG-3’ and 5′-GAGACCTCAGGAACATAATTG-3′ for RPS17 as housekeeping gene, respectively. Forward and reverse primers for pro-inflammatory cytokines were 5′-GCAACAAGTGGTGTTCTC-3′ and 5′-CAGATTCTTTTCCTTGAGGC-3′ (for IL-1B); 5′-GCAGAAAAAGGCAAAGAATC-3′ and 5′-CTACATTTGCCGAAGAGC-3′ (for IL-6); and 5′-CCATGTTGTAGCAAACCC-3′ and 5′-GAGTAGATGAGGTACAGGC-3′ (for TNF-a). β-Actin was used as housekeeping gene in the last three reactions. The forward and reverse primers were 5′-TCCTTCCTGGGCATGGAG-3′ and 5′-AGGAGGAGCAATGATCTTGATCTT-3′, respectively. RT-PCR was carried out on the Bio-Rad iQ5 system (Alcobendas, Madrid, Spain). The relative mRNA levels of the *TBK1* gene were normalized with the expression of RPS17 or β-Actin (2^−(∆CT gene of interest-∆CT RPS17 or β-Actin)^).

### 4.5. Immunoblotting Analysis

Cells were collected by centrifugation, washed with PBS, and total protein extracts were obtained by lysing them as previously described [39]. Nuclear and cytosolic fractions were obtained using the Subcellular Protein Fractionation Kit (Thermo Fisher Scientific, Madrid, Spain), following the manufacturer’s instructions. Lamin B and α-Tubulin were used as markers for nuclear and cytosolic fractions, respectively. The protein content of the extracts was determined by the Pierce BCA Protein Assay kit (Thermo Fisher Scientific, Madrid, Spain). Equal amounts of proteins were resolved by SDS–polyacrylamide gel electrophoresis. The proteins were then transferred to polyvinylidene fluoride (PVDF) membranes and immunodetected, as previously described [20]. The following primary antibodies were used: TDP-43 (1:1000); phosphor-(S409/410)-TDP-43 (1:500); β-actin (1:500); α-tubulin (1:1000) and Lamin B1 (1:1000), SQSTM/p62 (1:5000); phosphor-Ser403 p62; (1:1000); TBK1 (1:1000 TBK1 (1:1000), phosphor-Ser172 TBK1 (1:500); RIPK1 (1:2000); phosphor Ser161-RIPK1 (1:1000). Signals from the primary antibodies were amplified using species-specific antisera conjugated with horseradish peroxidase (Bio-Rad, Alcobendas, Madrid, Spain) and detected with a chemiluminescent substrate detection system, ECL. Relative band intensities were quantified using the ChemiDoc Imaging System (Bio-Rad Laboratories, Alcobendas, Madrid, Spain).

To study the autophagy flux, cells were treated with 30 μg/L hydroxychloroquine (HCQ) for the last 3 h of incubation before lysing. Proteins were extracted with 200 μL of lysis buffer (50 mM Tris-HCl pH 6.8, 10% glycerol and 2% sodium dodecyl sulfate (SDS) with protease inhibitors (Sigma-Aldrich, Madrid, Spain)) and phosphatase inhibitors (1 mM sodium orthovanadate decahydrate (Sigma-Aldrich, Madrid, Spain)). An amount of 20 μg of protein was loaded in CriterionTM TGX Precast MIDI Protein gels (Bio-Rad, Alcobendas, Madrid, Spain) and transferred to PVDF membranes (Bio-Rad, Alcobendas, Madrid, Spain).

### 4.6. Immunofluorescence

Lymphoblasts (1 × 10^6^ cells × mL^−1^) were fixed for 30 min in 4% paraformaldehyde in PBS and blocked and permeabilized with 0.5% Triton X-100 in PBS–0.5% BSA for 60 min at room temperature. Lymphoblasts were attached to poly-L-lysine-coated coverslips using a Cytospin centrifuge at 700 rpm for 7 min. Then, cells were incubated overnight with anti-TDP43 polyclonal antibody. After removing the primary antibodies, cells were washed with PBS and incubated with Alexa Fluor 488-conjugated anti-rabbit antibody. For nuclear staining, the preparations were mounted on ProLong^®^ Gold Antifade Reagent with DAPI (Thermo Fisher, Madrid, Spain) allowing nuclear visualization. High-resolution images were acquired for 45 cells per group in n = 3 independent experiments using the LEICA TCS-SP5-AOBS confocal microscope system (Heidelberg, Germany). Quantification of TDP-43 was performed using Image J software (version 1.53K). Data are expressed as the ratio of the fluorescence intensity of cytosolic TDP-43 vs. the intensity of the fluorescence of the nuclear protein.

### 4.7. Determination of Reactive Oxygen Species (ROS)

Intracellular accumulation of ROS was determined using the fluorescent probe CM-H2DCFDA (5-(and-6)chloromethyl 2′,7′-dichlorodihydrofluorescein diacetate, acetyl ester) (Invitrogen, C6827). Cells were loaded with 10 μM CM-H2DCFDA for 30 min. Fluorescence measurements were carried out using a Varioskan Flash (Thermoscientific Laboratories, Rockford, IL, USA). The CM-H2DCFDA probe had an excitation peak at a wavelength of 495 nm and an emission at a length of 510 nm.

### 4.8. Statistical Analysis

Statistical analyses were performed using Graph Pad Prism Software Version 8. Data are presented as means ± standard error of the mean (SEM). Statistical significance was estimated using the Student’s *t*-test or by analysis of variance (ANOVA) followed by the Fisher’s LSD test for multiple comparisons. We considered results to be significant at *p* < 0.05.

## 5. Conclusions

Here, we have investigated the effects of a missense mutation located in the dimerization domain of TBK1 on the protein functionality of lymphoblasts derived from an ALS patient. We have demonstrated that the ALS-associated R473 *TBK1* mutation inhibits p62 and RIPK1 phosphorylation, leading to an RIPK1-mediated increase in IL-6 levels without altering the autophagic flux. In addition, impaired TDP-43 homeostasis, enhanced ROS generation, and decreased cell viability were observed in mutant cells in contrast with cells derived from control individuals. Therapeutic strategies directed to reduce RIPK1 activity and to recover TDP-43 homeostasis could be useful for ALS patients with loss of or reduced TBK1 function mutations.

## Figures and Tables

**Figure 1 ijms-24-02847-f001:**
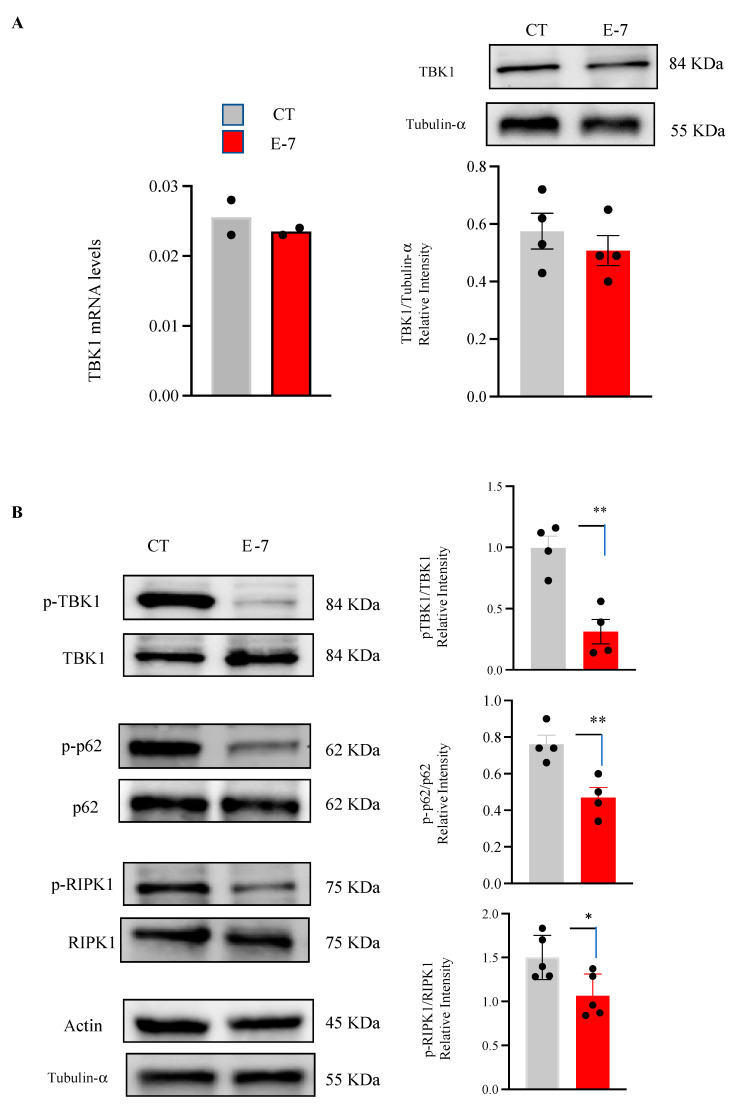
Lymphoblasts from pooled control samples and from the E7 patient were incubated at the initial density of 1 × 10^6^ cells × mL^−1^ in RPMI medium containing 10% FBS for 24 h. Then, cells were harvested to isolate RNA or to prepare extracts for WB. (**A**) Relative TBK1 mRNA and protein levels, determined in two and four independent experiments, respectively. (**B**) Effects of R573G variant on phosphorylated levels of TBK1 substrates. The data represent the means ± SEM of four or five different experiments. * *p* < 0.05, ** *p* < 0.01 significantly different from control cells.

**Figure 2 ijms-24-02847-f002:**
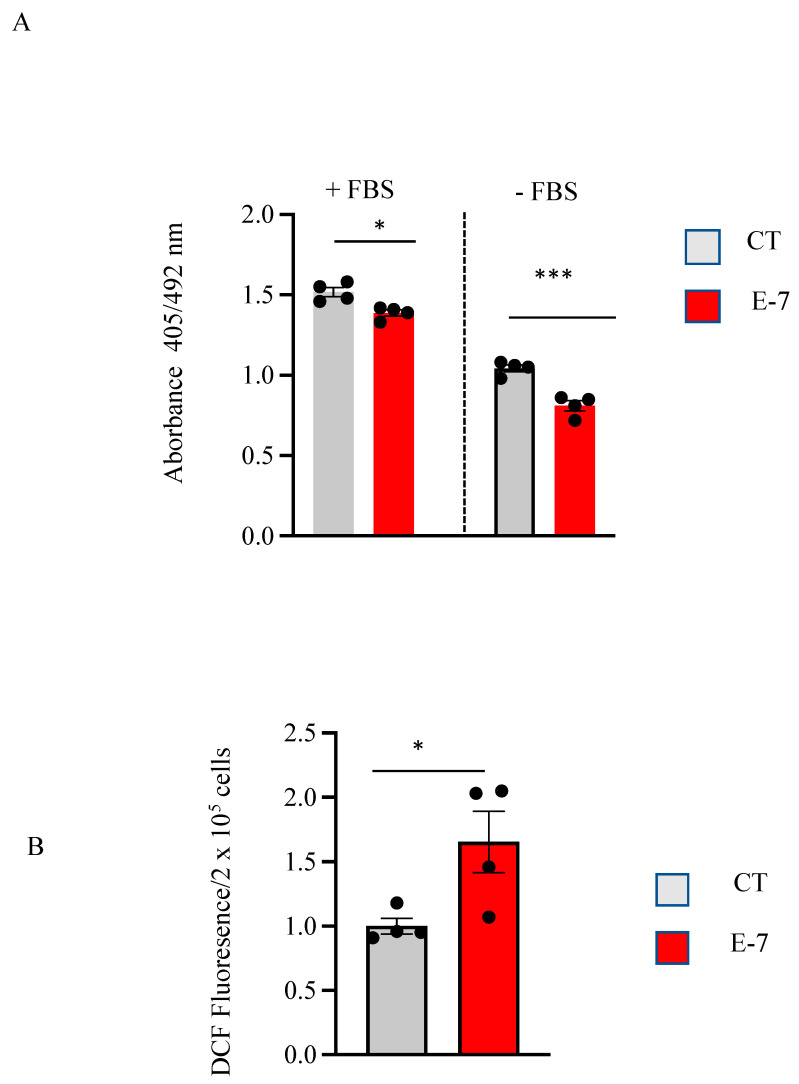
Effects of R563G *TBK1* mutation on cell viability and ROS generation. (**A**) Lymphoblasts from pooled control cells and from E7 ALS patient were seeded at an initial density of 1 × 10^6^ cells × mL^−1^ in the absence or presence of 10% FBS for 24 h. Cell viability was assessed by the MTT assay. The data represent the means ± SEM of four independent experiments carried out in triplicate. (**B**) Cells were incubated in RPMI medium containing 10% FBS for 24 h. Subsequently, about 200,000 cells were incubated with the CM-H2DCFDA probe for 30 min. The data show the means ± SEM of four independent experiments. * *p* < 0.05, *** *p* < 0.001, significantly different from control cells.

**Figure 3 ijms-24-02847-f003:**
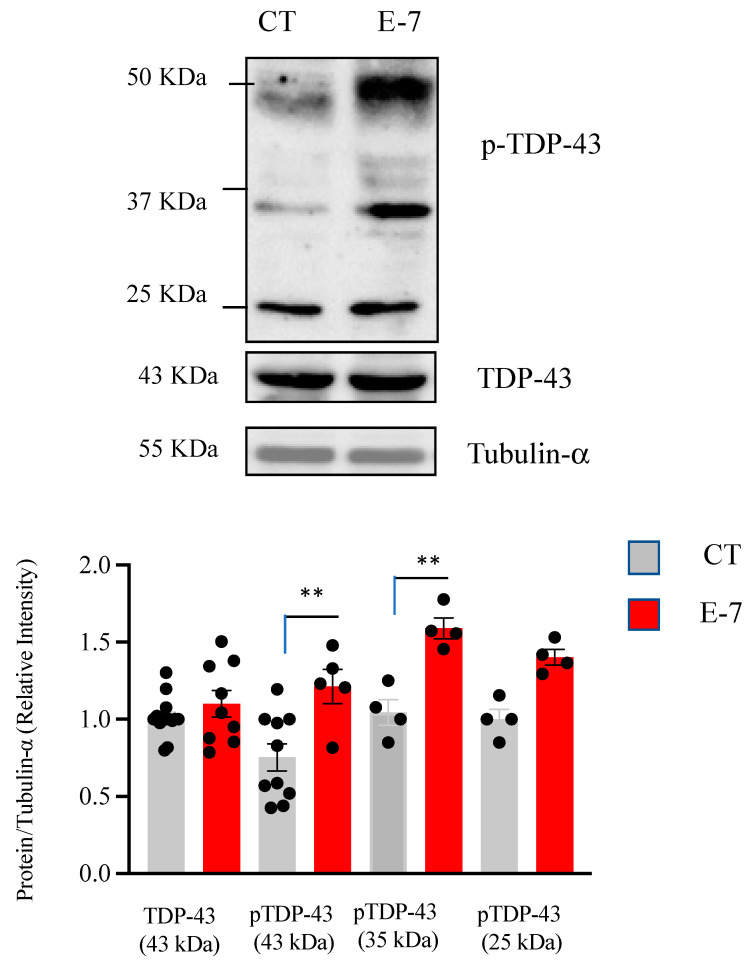
Cellular content and phosphorylation status of TDP-43 in lymphoblasts from control and the E7 patient. Lymphoblasts from pooled control cell lines and from the E7 patient were seeded at an initial density of 1 × 10^6^ cells × mL^−1^ in RPMI containing 10% FBS. After 24 h, cells were collected and processed to detect phospho-TDP-43 and total TDP-43 protein levels by Western blotting. Representative immunoblots are shown in upper panel. Tubulin-α was used as the loading control. Levels of phospho-TDP-43 (full length and truncated) and total TDP-43 are presented in lower panel. The data represent means ± SEM of four to ten independent experiments. ** *p* < 0.01, significantly different from control cells.

**Figure 4 ijms-24-02847-f004:**
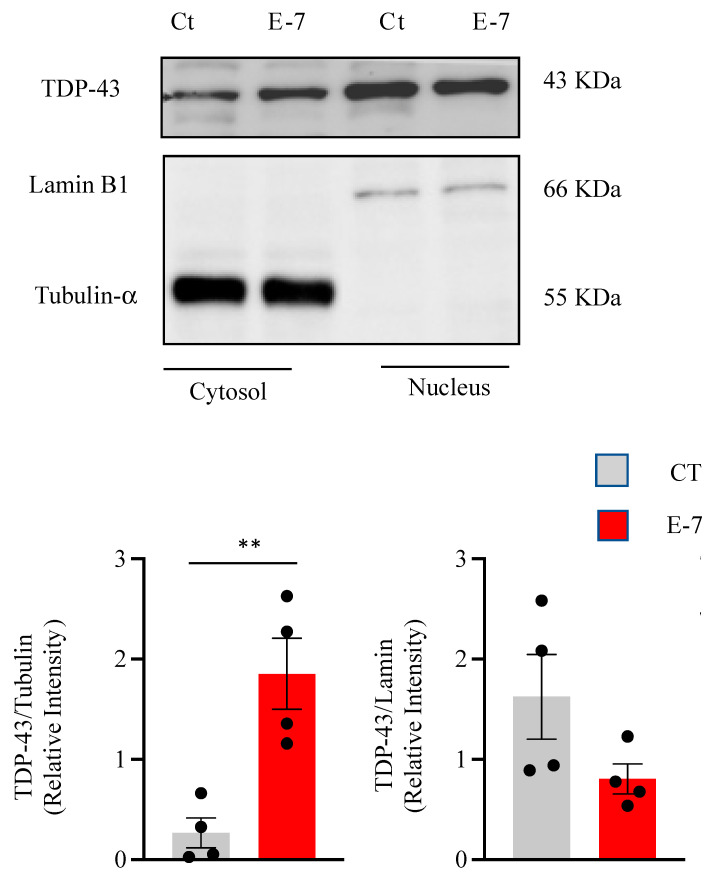
Subcellular localization of TDP-43 in control and E7 patient lymphoblasts. Lymphoblasts from pooled control cell lines and the E7 patient were seeded and incubated as in Figure 1. After harvesting, lymphoblasts were lysed to obtain cytosolic and nuclear fragments that were analyzed by Western blotting. Tubulin-α and Lamin B1 antibodies were used as loading and purity controls of the cytosolic and nuclear fractions, respectively. A representative experiment is shown. Densitometric analyses, shown below, represent the means ± SEM of four independent experiments. ** *p* < 0.01, significantly different from control cells.

**Figure 5 ijms-24-02847-f005:**
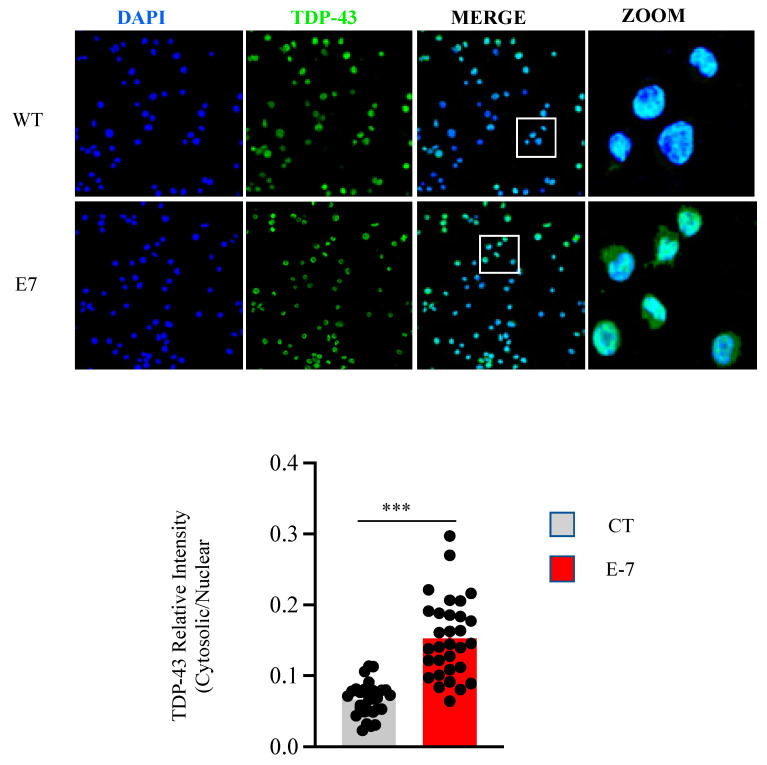
Confocal microscopy analysis on subcellular location of TDP-43 in control and E7 patient lymphoblasts. Lymphoblasts from pooled control cell lines and the E7 patient were seeded and incubated as in Figure 1. Confocal laser scanning microscopy was used to examine TDP-43 protein localization. DAPI was used to stain the cell nucleus. The relative fluorescence intensities of TDP-43 inside and outside the nuclei were determined in at least 35 cells per individual. The plotted values represent the means ± SEM of quantitative analyses of TDP-43 redistribution. Each dot represents the fluorescence intensity of a single cell. *** *p* < 0.001, significantly different from control cells.

**Figure 6 ijms-24-02847-f006:**
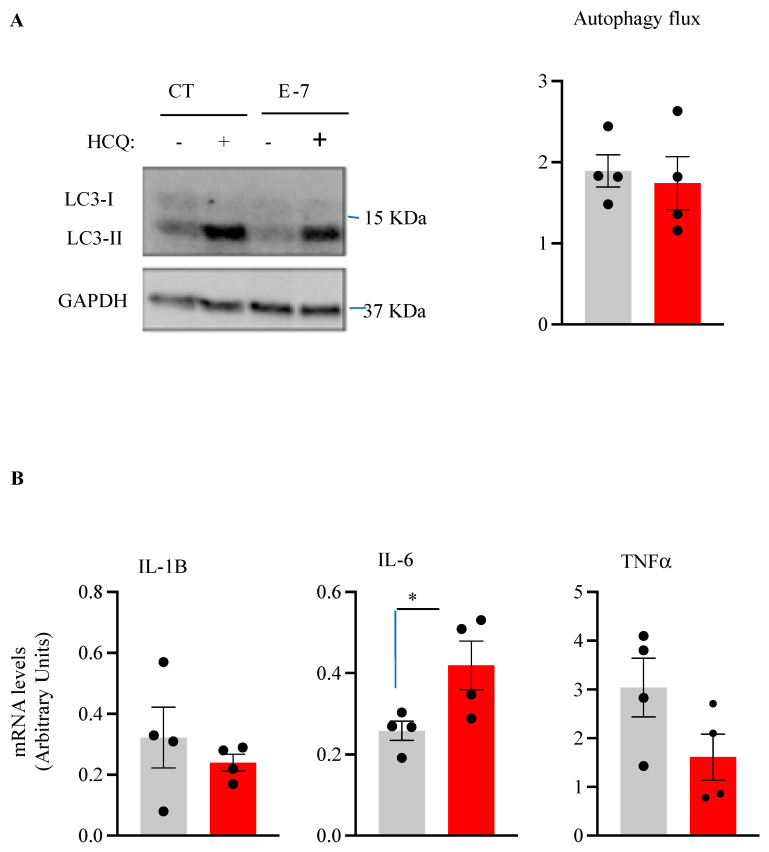
Influence of R563G *TBK1* mutation on autophagy flux and expression levels of pro-inflammatory cytokines. (**A**) Lymphoblasts from pooled control cell lines and from the E7 patient were seeded at an initial density of 1 × 10^6^ cells × mL^−1^ in RPMI containing 10% FBS for 24 h. Hydroxychloroquine (HCQ) was added during the last 3 h when indicated. Each point represents the mean of four independent experiments. (**B**) Quantitative real-time PCR determination of messenger RNA levels of pro-inflammatory cytokines IL-1, IL-6, and TNF-α, normalized by β-Actin messenger RNA levels. The data represent means ± SEM of four independent experiments. * *p* < 0.05, significantly different from control cells.

**Table 1 ijms-24-02847-t001:** Demographic and clinical characteristics of participants.

Code	Gender	Age	Clinical Presentation	Motor Neuron Affected
C105	Female	54	NA ^1^	NA
C106	Female	57	NA	NA
C112	Male	71	NA	NA
C126	Male	73	NA	NA
E7	Male	69	Spinal	MNS + MSI ^2^

^1^ NA: not applicable. ^2^ MNS, MNI: superior and inferior motor neuron, respectively.

## Data Availability

The datasets generated during the current study are available from the corresponding author upon reasonable request.

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
