# Peer review of "Functional Characterization of a Familial ALS-Associated Missense TBK1 (p-Arg573Gly) Mutation in Patient-Derived Lymphoblasts"

_ijms, 2023, doi:10.3390/ijms24032847_

Round 1

Reviewer 1 Report

This is a nice manuscript showing the functional effect of a missense mutation in the ALS-associated TBK1 gene. Authors show a decrease in TBK1 phosphorylation together with a reduced phosphorylation of its substrates p62 and RIPK1, TDP-43 cellular pathology, higher ROS and IL-6 levels. Authors didn’t see altered aytoohagic flux or increased cell death. The experiments are technically well executed and the results are sound.

Minor comments 

What are the p-values and statistical test used in fig 2 A? There seems to be a clear difference between ctr and patient specially in the absence of FBS.

Fig 6 A. Please correct. I assume left panel is control with/without HCQ and right is the patient. Also check figure legend; does each point represent the mean of 4 experiments? Please explain in the axis how is the autophagic flux calculated.

The paper is clear and well written, but there are some typos/gramatical errors throughout the manuscript, please check (e.g. (but not only) lines 3, 114, 158, 173, 187, 340, 344, 365)

Author Response

Manuscript ID: ijms-2136075

RESPONSE TO REVIEWER #1

This is a nice manuscript showing the functional effect of a missense mutation in the ALS-associated TBK1 gene. Authors show a decrease in TBK1 phosphorylation together with a reduced phosphorylation of its substrates p62 and RIPK1, TDP-43 cellular pathology, higher ROS and IL-6 levels. Authors didn’t see altered aytoohagic flux or increased cell death. The experiments are technically well executed and the results are sound.

We thank Reviewer 1 for his/her kind comments

Minor comments 

What are the p-values and statistical test used in fig 2 A? There seems to be a clear difference between ctr and patient specially in the absence of FBS.

We thank Reviewer #1 for this comment. After revising the experimental data, we noticed that there was a mistake in the statistical analysis. Indeed there is a significant decrease in cell viability of patient-derived lymphoblasts, compared with that observed in control cells. The statistical significance has been added to the new Fig. 2A. and the corresponding paragraph in the Results section has been corrected

Fig 6 A. Please correct. I assume left panel is control with/without HCQ and right is the patient. Also check figure legend; does each point represent the mean of 4 experiments? Please explain in the axis how is the autophagic flux calculated.

Reviewer #1 is right, therefore fig. 5A has been corrected. As suggested, autophagic flux is now defined in the Y axis of Fig. 5A. as the ratio of relative intensities of LC3-II in the presence of HCQ and in the absence of HCQ. Autophagic flux was determined in pooled control cells and E7 lymphoblasts in four independent experiments.

The paper is clear and well written, but there are some typos/gramatical errors throughout the manuscript, please check (e.g. (but not only) lines 3, 114, 158, 173, 187, 340, 344, 365)

All typos errors have been corrected

Reviewer 2 Report

Porras et al described in the manuscript the characterization of a missense mutation in the TBK1 gene which is linked to ALS. Therefore, they immortalized lymphocytes from one patient with the missense mutation and compared that line to 4 control lines. The cell line was characterized for the TBK1 expression, the expression of known interaction partner, TDP-43 expression and accumulation as well as autophagic flux. In total, the manuscript is well written and demonstrate important results. I have a few major comments and some minor comments.

Major comments:

(I) In the main manuscript the authors used pooled control samples for western blot and confirmed once for figure 1 that the controls showed similar expression for TBK in supplementary figure 1.  That’s absolutely fine. But checking supplementary information (original images/blots) showed directly on the first blot (Mbr P-TBK/NAK) that the different controls differ in their P-TBK expression very much. There, the pooled control samples showed a very high amount of P-TBK, but the two shown single controls showed nearly no expression of P-TBK and therefore, in total less P-TBK as the ALS cell line. But the author concluded in the main manuscript that in the ALS cell line less P-TBK is expressed compared to pooled controls. What is the explanation that the two shown controls have less but the pooled controls more P-TBK and how did behave the other two controls. Maybe here it is not possible to pool the controls and two shown only single results for each control.

(II) For the western blots shown in figure 1B, the quantification demonstrated a ratio of total vs phospho-protein expression. Unclear is if these ratios were normalized before to the housekeeping proteins or not. Because it would be important to normalized first to the respective housekeeping proteins before calculating the ratio as the housekeeping proteins demonstrate a slight loading differences. It should be also mentioned in a better way how quantification was performed. Additionally, the author mentioned in the figure legend for 1B that they analyzed four different experiments but up to five single dots are shown in the quantification. That similar for other quantifications in other figures.

(III) It is unclear how the autophagic flux was calculated. Did the author also took into account the LC3 I to LC3 II ratio and the GAPDH expression? E-7 express much more LC3 in the present Western blot, but also only E-7 was treated with HCQ as indicated in the blot. For comparable reasons it would be important to also stress the wildtype cells with HCQ and not only the patient cells.

(IV) Expression of interleukins is only shown on RNA level. To say something about the involvement of interleukins it would be important to also measure protein levels by western blot or ELISA.

Minor comments:

A) The western blots in the main manuscript are not labeled properly, there is no kDa labeling, therefore unclear which size the detected proteins have. Western Blot figure 1a is also not labeled with the genotype.

(II) The stars which indicate significance are hard to discriminate from sample circles. Also, in the main results the p-values should be indicated.

(III) The number how often the experiments are repeated are unclear. Did each dot in the quantification represents one single experiment or also the duplicates/triplicates within the same experiment? Please indicate in each figure legend clearly how many experiments and how many triplicates and what is shown exactly in the quantification. Additionally, in figure 5 did the quantification show the number of cells quantified or the number experiments, because there is a high amount of single dots and unclear how the quantification was performed.

(IV) The explanation for full-length, truncated form of TDP-43 is quite short and not really understandable. Please indicated in the main result section information of the kDa size of full-length, truncated TDP-43 corresponding to what is seen in blot 3.

(V) There is no scale bar shown in figure 5 and also not indicated that in the last picture a higher magnification is shown.

(VI) The raw western blot data are not probably labeled, sometimes the name is missing or two names are on one single lane. In some pictures the marker is labeled in other not. Or the indicated marker is not the same as in other blots where the same blue/red lanes are labeled with different kDa size in different blots. Also, the sample names are completely unclear what is the difference between e.g. PCt1, PCt11,  PCT10, E7-1, E7-4

(VII) There are some spelling errors e.g. line 158, 169, 178, 185, 198….. which need to be corrected.

(VIII) Is it really true that for autophagic flux in total 20 g protein where loaded as indicated in the method section?

Author Response

Manuscript ID: ijms-2136075

RESPONSE TO REVIEWER #2

Porras et al described in the manuscript the characterization of a missense mutation in the TBK1 gene which is linked to ALS. Therefore, they immortalized lymphocytes from one patient with the missense mutation and compared that line to 4 control lines. The cell line was characterized for the TBK1 expression, the expression of known interaction partner, TDP-43 expression and accumulation as well as autophagic flux. In total, the manuscript is well written and demonstrate important results. I have a few major comments and some minor comments.

We are grateful to the reviewer for his/her insightful comments.  We think that by addressing the issues raised, the manuscript has been greatly improved

Major comments:

(I) In the main manuscript the authors used pooled control samples for western blot and confirmed once for figure 1 that the controls showed similar expression for TBK in supplementary figure 1.  That’s absolutely fine. But checking supplementary information (original images/blots) showed directly on the first blot (Mbr P-TBK/NAK) that the different controls differ in their P-TBK expression very much. There, the pooled control samples showed a very high amount of P-TBK, but the two shown single controls showed nearly no expression of P-TBK and therefore, in total less P-TBK as the ALS cell line. But the author concluded in the main manuscript that in the ALS cell line less P-TBK is expressed compared to pooled controls. What is the explanation that the two shown controls have less but the pooled controls more P-TBK and how did behave the other two controls. Maybe here it is not possible to pool the controls and two shown only single results for each control.

We greatly appreciate this reviewer’s observation. As indicated in Supplementary Fig 1 the levels of phospho-TBK1, and TBK1 were determined in 4 control cell lines and compared with pooled samples, finding no differences. The experiment was repeated twice. For the sake of clarity, the former original blot has been cut off to avoid confusion. We believe that, in this particular experiment something went wrong with extracts prepared from cell lines C106 and C112.  In the new “original blots” file, additional data is now provided in which extracts from pooled control and E7 cell lines extracts, obtained in 4 independent experiments, were used to analyze the status of phosphorylation of TBK1.

 (II) For the western blots shown in figure 1B, the quantification demonstrated a ratio of total vs phospho-protein expression. Unclear is if these ratios were normalized before to the housekeeping proteins or not. Because it would be important to normalized first to the respective housekeeping proteins before calculating the ratio as the housekeeping proteins demonstrate a slight loading differences. It should be also mentioned in a better way how quantification was performed. Additionally, the author mentioned in the figure legend for 1B that they analyzed four different experiments but up to five single dots are shown in the quantification. That similar for other quantifications in other figures

Relative intensities for each protein listed in Fig. 1B were first normalizes for a housekeeping protein before the determination of phosphorylated versus total protein. Legend for all figures have been checking to indicate the exact number of experiments preformed.

(III) It is unclear how the autophagic flux was calculated. Did the author also took into account the LC3 I to LC3 II ratio and the GAPDH expression? E-7 express much more LC3 in the present Western blot, but also only E-7 was treated with HCQ as indicated in the blot. For comparable reasons it would be important to also stress the wildtype cells with HCQ and not only the patient cells.

We apologize for the mislabeling of former Fig. 6A. .LC3-I and LC3-II were determined on both pooled control cells and E7 lymphoblasts in the absence and in the presence of HCQ. Autophagic flux was then calculated as the ratio of relative intensity of LC·-II (normalize by that of GAPDH,) with HCQ, versus LC3-II /GAPDH without HCQ. New Fig 6 has been corrected to clarify this point

(IV) Expression of interleukins is only shown on RNA level. To say something about the involvement of interleukins it would be important to also measure protein levels by western blot or ELISA.

Among the different methods available to determine cytokine levels, we choose the “real real-time quantitative polymerase chain reaction (Q-PCR), to measure cytokine mRNA transcript abundance, which correlates with protein levels. This method is relatively straightforward and quantitative and allows for the detection of many different cytokines from relatively small sample amounts. (Amsen et al (Amsen D, de Visser KE, Town T. Approaches to determine expression of inflammatory cytokines. Methods Mol Biol. 2009;511:107-142. doi:10.1007/978-1-59745-447-6_5)

Minor comments:

  1. A) The western blots in the main manuscript are not labeled properly, there is no kDa labeling, therefore unclear which size the detected proteins have. Western Blot figure 1a is also not labeled with the genotype.

All figures have been corrected in the revised Ms, to include KDa information for each protein

(II) The stars which indicate significance are hard to discriminate from sample circles. Also, in the main results the p-values should be indicated.

 The reviewer is right, therefore we have modified all figures to clarify this issue.

(III) The number how often the experiments are repeated are unclear. Did each dot in the quantification represents one single experiment or also the duplicates/triplicates within the same experiment? Please indicate in each figure legend clearly how many experiments and how many triplicates and what is shown exactly in the quantification. Additionally, in figure 5 did the quantification show the number of cells quantified or the number experiments, because there is a high amount of single dots and unclear how the quantification was performed.

We have checked all legend to Figures to indicate the precise number of experiments preformed. Each dot on Figs 1, 2,3, 4, and 6 represents an independent experiment carried out in control and patient E7-derived lymphoblasts. Regarding Fig. 5, Relative fluorescence intensity of TDP-43 inside and outside nuclei was determined in at least 35 cells per individual.  The experiment was repeated three times. Cytosolic and nuclear TDP-43 levels per cell were quantified using the Image J software (version 1.53K).

 (IV) The explanation for full-length, truncated form of TDP-43 is quite short and not really understandable. Please indicated in the main result section information of the kDa size of full-length, truncated TDP-43 corresponding to what is seen in blot 3.

Done

(V) There is no scale bar shown in figure 5 and also not indicated that in the last picture a higher magnification is shown.

It is now indicated in new Fig, 5 , that last pictures in each row  correspond to higher magnification-

(VI) The raw western blot data are not probably labeled, sometimes the name is missing or two names are on one single lane. In some pictures the marker is labeled in other not. Or the indicated marker is not the same as in other blots where the same blue/red lanes are labeled with different kDa size in different blots. Also, the sample names are completely unclear what is the difference between e.g. PCt1, PCt11, PCT10, E7-1, E7-4

We apologize for the labelling of the raw bots. The differences between e.g. PCt1, PCt11 refer to identical pooled samples prepared in different occasions. to avoid confusion, the blots had been labelling only with Pct and E7.

(VII) There are some spelling errors e.g. line 158, 169, 178, 185, 198….. which need to be corrected.

All spelling errors have been corrected

 (VIII) Is it really true that for autophagic flux in total 20 g protein where loaded as indicated in the method section?

We thank Reviewer 2 for this observation. The amount of protein loaded to determine autophagic flux was 20µg

Round 2

Reviewer 2 Report

The author addressed all points properly.